# INHERENTLY INTERPRETABLE TREE ENSEMBLE LEARNING

## ABSTRACT

Tree ensembles such as random forests and gradient boosting machines are among the most effective methods for tabular prediction, but their strong performance often comes at the cost of interpretability. We show that ensembles of shallow decision trees admit an equivalent functional ANOVA representation, making them inherently interpretable while retaining competitive accuracy. Building on this insight, we develop an exact algorithm that decomposes tree ensembles into main effects and interactions, yielding faithful explanations without approximation. We further introduce two strategies to enhance interpretability: (i) imposing constraints on depth, monotonicity, and interactions, and (ii) post-hoc pruning of trivial effects via sparse modeling and effect selection. Across synthetic and real-world datasets, our approach achieves a superior trade-off between interpretability and predictive power compared to established interpretable models such as Explainable Boosting Machines and GAMI-Net. These results position shallow tree ensembles as a practical and theoretically grounded alternative for transparent high-performance modeling of tabular data.

## 1 INTRODUCTION

Tree ensembles are widely recognized as one of the most popular machine learning techniques for modeling tabular data. For example, a bagging tree aggregates multiple regression or classification trees by making bootstrap replicates of the training data (Breiman, 1996). The random forest also averages a bunch of decision trees to reduce the variance, and it combines the bagging and random feature selection strategies to draw training samples for every single tree (Breiman, 2001). In contrast to constructing trees independently, gradient-boosted machines employ a sequential fitting approach. Each new tree in the ensemble is added to address the deficiencies of the previous trees and enhance the model's performance (Friedman, 2001).

In general, gradient-boosting trees tend to exhibit superior predictive performance compared to random forests and bagging trees. The state-of-the-art implementations of gradient-boosted machines include XGBoost (Chen & Guestrin, 2016), LightGBM (Ke et al., 2017), and CatBoost (Dorogush et al., 2018), in which they have developed a wide range of extensions and enhancements built upon the naïve algorithm. Although tree ensemble models demonstrate superb predictive performance, they often suffer from the model interpretation challenge. A well-performing tree ensemble model usually consists of a large number of trees. Each tree can be interpreted separately, but it becomes almost impossible to understand and interpret the whole model. As a result, tree ensemble models are usually perceived as black boxes.

Functional analysis of variance (ANOVA) (Stone, 1994; Huang, 1998) is a promising framework for interpreting black-box models. It decomposes a model as the sum of additive components. In this paper, we demonstrate that when shallow decision trees are used as base learners, tree ensemble models can not only become inherently interpretable but also sometimes lead to better generalization performance. The main contribution of this paper is the development of a practical pipeline for building models that are both high-performance and exactly interpretable, by leveraging the inherent structure of shallow tree ensembles, as summarized below.

- We demonstrate that shallow tree ensembles (depth 2) are functionally equivalent to a GA²M and provide an exact algorithm to decompose them. This allows for faithful, non-

approximate interpretation of a mainstream, high-performance model class, a significant advantage over model-agnostic approximation methods.

- We systematize the process into a coherent methodology: a) using standard hyperparameters to design an exactly-decomposable model, b) applying an exact transformation to reveal the GA²M structure, and c) providing a pruning strategy to distill the model into its most parsimonious form.

- We show that this approach achieves a superior performance-interpretability trade-off compared to specialized interpretable models (NAM, EBM, GAMI-Net). It provides a compelling, off-the-shelf alternative that delivers both the accuracy of tree ensembles and the exact, transparent explanations of a GA²M.

## 2 RELATED WORK

Interpretable machine learning techniques can be broadly categorized into post-hoc explanation tools and inherently interpretable models. Post-hoc tools such as PDP (Friedman, 2001), ALE (Apley & Zhu, 2020), LIME (Ribeiro et al., 2016), and SHAP (Lundberg & Lee, 2017; Lundberg et al., 2020) explain complex models after training but may produce approximations that deviate from the true model behavior (Rudin, 2019). Inherently interpretable models, on the other hand, are designed with constraints such as additivity, sparsity, and smoothness to ensure transparency without sacrificing accuracy (Sudjianto & Zhang, 2021). Classic examples include generalized additive models like EBM (Lou et al., 2013) and GAMI-Net (Yang et al., 2021), while recent advances such as NAMs (Agarwal et al., 2021), NODE-GAM (Chang et al., 2021), SPAM (Dubey et al., 2022), SIAN (Enouen & Liu, 2022), and Gamformer (Mueller et al., 2024) leverage neural networks and scalable architectures to capture complex feature relationships while maintaining interpretability. Furthermore, extensions such as Neural Additive Models for Location, Scale, and Shape (NAMLSS) (Thielmann et al., 2024) expand the GAM framework beyond modeling only the conditional mean, enabling interpretable modeling of distributional properties.

While neural GAMs can offer more flexible shape functions on large datasets, our method provides a practical and faithful interpretability solution for tree ensembles, with the additional advantage of potentially better generalization performance compared with EBM for shallow trees. A comprehensive review of related works is available in Appendix A.

## 3 PRELIMINARY AND NOTATIONS

**Tree Ensemble Models.** A tree ensemble model, such as XGBoost or LightGBM, can be represented as the addition of (tree, weight)-pairs

$$f(\mathbf{x}) = \sum_{k=1}^{K} w_k T_k(\boldsymbol{x}),  \tag{1}$$

where $K$ is the total number of trees. In gradient boosting, the weights $w_k$ correspond to the learning rates. Each tree $T_k$ can be further represented as the addition of leaf nodes. By rearranging the additive components, we can represent (1) as the addition of all leaf nodes, as follows,

$$f(\mathbf{x}) = \sum_{m=1}^{M} v_m \prod_{j \in S_m} I\left(s_{mj}^l \leq x_j < s_{mj}^u\right),  \tag{2}$$

where $M$ is the total number of leaf nodes and $v_m$ is the value of the $m$-th leaf node, multiplied by the corresponding tree weight. The symbol $S_m$ represents the set of split variables in the decision path of the $m$-th leaf node. The product of indicator functions denotes whether a sample belongs to the corresponding leaf node. In specific, the interval $[s_{mj}^l, s_{mj}^u)$ is determined by the following rules.

- If a tree has no split, then $s_{mj}^l = -\inf$ and $s_{mj}^u = \inf$. This is a special case where the root node stops splitting, and it corresponds to an intercept term.

- As a feature is used only once in the decision path, and the leaf node belongs to the left side of the split point $s$, then $s^u_{mj} = s$ and $s^l_{mj} = -\inf$. Otherwise, if the leaf node belongs to the right side of the split point, then $s^l_{mj} = s$ and $s^u_{mj} = \inf$.

- As a feature is used multiple times in the decision path, then $s^l_{mj}$ and $s^u_{mj}$ are determined by the intersection of these split-generated intervals.

**Functional ANOVA.** Functional ANOVA decomposes a model as the sum of additive components, as follows.

$$f(\mathbf{x}) = \mu + \sum_j f_j(x_j) + \sum_{jk} f_{jk}(x_j, x_k) + \dots, \tag{3}$$

where $\mu$ is the intercept, which captures the global mean, and

- The **main effect** $f_j(x_j)$ shows how the output changes as $x_j$ varies;
- The **pairwise interaction** $f_{jk}(x_j, x_k)$ measures how $x_j$ and $x_k$ **jointly influence** the prediction beyond what can be explained by their individual effects.

If $f_{jk} = 0$, the relationship between $x_j$ and $x_k$ is **additive**, meaning there is no interaction between them. Higher-order interactions capture interactions among three or more features. Each component is orthogonal to lower-order components and has zero mean under its respective variables.

## 4 METHOD

Our goal is to interpret the prediction function $f(\mathbf{x})$ using additive effects associated with different subsets of features, following a functional ANOVA-style decomposition in (3). Figure 1 shows the proposed pipeline of building inherently interpretable tree ensemble models by combining interpretability constraints, functional ANOVA representation, and post-hoc effect pruning.

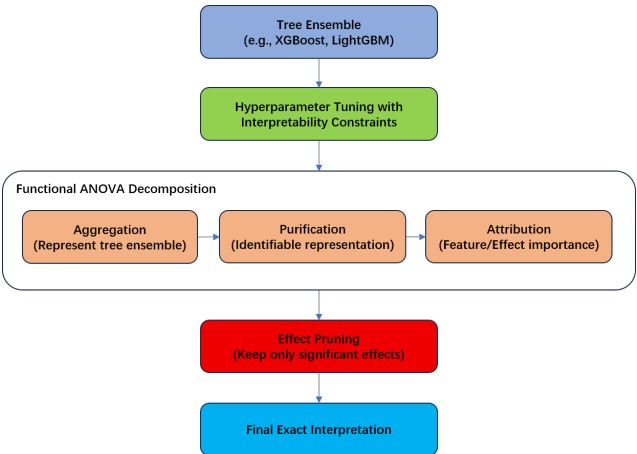

Figure 1: Pipeline of the tree ensemble interpretation framework.

### 4.1 TRAINING WITH INTERPRETABILITY-ORIENTED CONSTRAINTS

To enhance the interpretability of tree ensemble models like XGBoost, specific hyperparameters can be configured to control model complexity, enforce domain knowledge, and simplify the resulting functional ANOVA structure. Table 1 summarizes the most important hyperparameters and their interpretability roles. Detailed explanations and examples are provided in Appendix B.

By carefully configuring these hyperparameters, practitioners can balance predictive performance with interpretability, ensuring that the resulting tree ensemble models remain both accurate and transparent.

Table 1: Key hyperparameters for improving interpretability of tree ensemble models.

| Hyperparameter | Interpretability Effect |
|---|---|
| **Max Tree Depth** | Depth 1: only main effects (GAM); Depth 2: main + pairwise (EBM-like), etc. |
| **Monotonicity** | Prevents counterintuitive patterns (e.g., higher income leading to lower credit score). |
| **Max Bins** | Reduces unnecessary discontinuities, making effects smoother and easier to visualize. |
| **Interaction** | Keeps model focused on domain-relevant interactions. |
| **Regularization** | Produces sparser functional ANOVA representation by shrinking or eliminating insignificant effects. |
| **Early Stopping** | Prevents overfitting and curbs model growth, leading to simpler and more interpretable structures. |

## 4.2 REPRESENTING TREE ENSEMBLE MODELS VIA FUNCTIONAL ANOVA

Given a fitted tree ensemble, we can represent it using the functional ANOVA framework. The algorithm can be divided into three steps, i.e., aggregation, purification, and attribution.

### 4.2.1 AGGREGATION

The first step is to rearrange (2) using the functional ANOVA framework defined in (3), by assigning each leaf node to the effect functions. For each leaf node, its corresponding effect function is determined by the distinct split variables at its decision path. For example, leaf nodes with only one distinct split variable are the main effects. The $j$-th main effect $f_j(x_j)$ is obtained by the sum of all the leaf node functions subject to $S_m = \{j\}$, as follows,

$$f_j(x_j) = \sum_{S_m = \{j\}} v_m \cdot I\left(s_{mj}^l \leq x_j < s_{mj}^u\right). \tag{4}$$

Leaf nodes with two distinct split variables correspond to pairwise interactions. A pairwise interaction $f_{jk}(x_j, x_k)$ can be calculated by the sum of all the leaf nodes subject to $S_m = \{j, k\}$, as follows,

$$f_{jk}(x_j, x_k) = \sum_{S_m = \{j,k\}} v_m \cdot I\left(s_{mj}^l \leq x_j < s_{mj}^u\right) \cdot I\left(s_{mk}^l \leq x_k < s_{mk}^u\right). \tag{5}$$

Similarly, leaf nodes with more than two distinct split variables are assigned to the corresponding higher-order interaction terms. For a depth-$d$ tree ensemble model, each leaf node would have at most $d$ distinct split variables, and hence the highest possible interaction order is also $d$. In particular, a shallow tree ensemble with a maximum depth of 1 can be represented as a generalized additive model (GAM). A depth-2 tree ensemble can be represented as a generalized additive model with pairwise interaction (GAMI), etc.

Note that all effect functions are piece-wise constant, representing a weighted sum of indicator functions. A main effect $f_j(x_j)$ with $N_j$ distinct split points can be represented as a value vector of length $N_j + 1$. A pairwise interaction $f_{jk}(x_j, x_k)$ with $N_j$ and $N_k$ distinct split points on features $j$ and $k$, respectively, can be represented as a matrix of size $(N_j + 1, N_k + 1)$. In general, higher-order effects can also be represented using higher-order tensors, using a similar approach.

### 4.2.2 PURIFICATION

The functional ANOVA would suffer from the identifiability issue without any constraint. For example, a main effect term can be absorbed into its parent interactions without changing the model prediction. This will lead to multiple equivalent representations and make the interpretation non-unique. To ensure a unique interpretation, it is assumed that the decomposed effects satisfy the following constraint

$$\int f_{i_1 \cdots i_t}\left(x_{i_1}, \cdots, x_{i_t}\right) dx_k = 0, \quad k = i_1, \cdots, i_t, \tag{6}$$

where $i_1, \cdots, i_t$ are feature indices. It implies that all main and interaction effects a) have zero means and b) are mutually orthogonal, i.e.,

$$\int f_{i_1 \cdots i_u}\left(x_{i_1}, \cdots, x_{i_u}\right) f_{j_1 \cdots j_v}\left(x_{j_1}, \cdots, x_{j_v}\right) d\mathbf{x} = 0, \tag{7}$$

whenever $(i_1, \cdots, i_u) \neq (j_1, \cdots, j_v)$.

In the aggregation step, we have rearranged all the leaf node rules to the corresponding effects. However, these raw effects do not necessarily satisfy the functional ANOVA constraint in (6). To address this issue, an effective purification algorithm proposed by Lengerich et al. (2020) is applied. For an arbitrary effect $f_{i_1 \cdots i_t}(x_{i_1}, \cdots, x_{i_t})$, it approximates (6) by removing the means of each slice feature $i_1 \cdots i_t$ iteratively and sequentially. The removed effects are then added to the corresponding child effects to ensure the equivalence of the purified model and the original model.

For simplicity, we illustrate this algorithm using a pairwise interaction $f_{jk}(x_j, x_k)$. We take the matrix representation of the pairwise interaction (of size $(N_j + 1, N_k + 1)$) as input. This algorithm then operates on the matrix using the following steps:

- Calculate the average value along the first dimension, and get a mean vector of size $(N_k + 1)$. Subtract the mean vector from the value matrix, and add it to the corresponding main effect $f_k(x_k)$.

- Calculate the average value along the second dimension, and get the mean vector of size $(N_j + 1)$. Subtract the mean vector from the value matrix, and add it to the corresponding main effect $f_j(x_j)$.

These two steps are repeated multiple times until convergence, i.e., as the maximum absolute difference of the matrix between two consecutive iterations is less than a predefined threshold. In the end, we would get a purified pairwise interaction, as well as two updated child main effects. In general, for a $d$-way interaction, the purification algorithm would iterate over each dimension of the corresponding $d$-way tensor, and for each dimension, it moves the $(d - 1)$-way mean tensor to the corresponding child $(d - 1)$-way interaction. The final result would be a purified $d$-way interaction, together with $d$ child $(d - 1)$-way interactions.

The whole purification algorithm would start from the highest-order interactions and recursively cascade effects from high-order interactions to low-order interactions. Finally, for main effects, we can simply center them to have zero means, and the subtracted mean is then added to the intercept term. As the purification step finishes, we can visualize the main effects through 1D line plots and pairwise interactions via 2D heatmaps. For higher-order interactions, we can draw 1D or 2D plots for one or two features of interest, while fixing the rest features to certain representative values.

**Complexity analysis.** The purification algorithm becomes increasingly expensive as the interaction order $d$ grows. Both time and memory scale exponentially with $d$ because each purification iteration requires $O(d \cdot N^d)$ time $O(N^d)$ memory, where $N$ is the number of bins per feature. This makes it feasible for main effects ($d = 1$) and pairwise interactions ($d = 2$), and possibly $d = 3$ with small $N$. However, for $d > 3$, the computational and storage requirements quickly become prohibitive, so in practice, purification is typically limited to low-order interactions.

In the above discussion, we assume the data is uniformly and independently distributed over the feature space, which may not be the case in practical applications. The weighted functional ANOVA decomposition (Hooker, 2007) is accordingly proposed by considering the empirical distribution of data. To use weighted functional ANOVA, we first calculate the probability for each bin of the matrix / tensor, and the simple average is replaced by the weighted average.

### 4.2.3 ATTRIBUTION

As we have converted a tree ensemble model into the functional ANOVA representation, the next step is to quantify the contribution of the decomposed effects, both locally (for an individual sample) and globally (for the entire dataset). Below, we introduce the definition of effect contributions and feature contributions.

The effect-level contribution quantifies the contribution of each effect to the model output. For example, the contribution of the $j$-th main effect is $f_j(x_j)$, and $f_{jk}(x_j, x_k)$ is the contribution of the pairwise interaction $(j, k)$, etc.

**Local effect contribution.** The model output for each sample can be interpreted as the sum of all effect contributions plus the intercept term. Each effect contribution can have a positive, negative, or

zero value. By considering the magnitude of effect values, we can select the most significant effects for an individual sample.

**Global effect importance.** After calculating the effect contributions for each sample, we can summarize the importance of each effect by examining the variance of the local effect contributions across a given dataset, such as the training data. Subsequently, the effect's importance is normalized in a way that ensures the sum of all effects' importance equals 1.

In contrast, the feature-level contribution quantifies the contribution of a feature $j$ to the model output of an individual sample, i.e.,

$$z_j(x_j) = f_j(x_j) + \frac{1}{2}\sum_k f_{jk}(x_j, x_k) + \frac{1}{3}\sum_{kl} f_{jkl}(x_j, x_k, x_l) + \cdots + \frac{1}{p}f_{1\cdots p}(x_1, \cdots, x_p). \quad (8)$$

In this formula, the main effect $f_j(x_j)$ is added directly to the $j$-th feature contribution. Additionally, all pairwise interaction effects associated with feature $j$ are included in the feature contribution, but with a discount factor of 2. This rule is also extended to 3-way, 4-way, and up to $p$-way interactions, where $p$ is the number of features. Note that the feature contribution $z_j(x_j)$ is derived from the Shapley value (Shapley, 1953) of feature $j$, defined as follows,

$$\phi_j = \sum_{S \subseteq \{1,\ldots,p\}\setminus\{j\}} \frac{|S|!(p - |S| - 1)!}{p!} \left(v(S \cup \{j\}) - v(S)\right), \quad (9)$$

where $v$ is the value function that returns the prediction of each feature coalition $S$. The marginal contribution of feature $j$ to the coalition $S$ is quantified by $v(S \cup \{j\}) - v(S)$, and the multiplier on the left is the weight of feature coalitions. In the functional ANOVA framework, the value function of different feature coalitions is already defined. The proof of the equivalence between Shapley value and feature contribution $z_j(x_j)$ can be found in Owen (2014). In shallow tree ensemble models, we can exactly calculate the Shapley value / feature contribution without much computational burden. According to $z_j(x_j)$, we define the following feature-level importance.

**Local feature contribution.** Similar to the local effect contribution, we can locally interpret the model output of an individual sample at the feature level, i.e., by $z_j(x_j)$.

**Global feature importance.** The significance of feature $j$ is determined by evaluating the variance of $z_j(x_j)$ on a specific dataset, such as the training data. After that, we normalize the feature importance to ensure that the total importance of all features adds up to 1.

### 4.3 Pruning Trivial Effects for Concise Interpretations

To enhance the interpretability of a tree ensemble model, we can prune trivial effects after it is fitted. This can be approached as a supervised feature selection problem, where each effect is treated as a feature. Various existing feature selection algorithms can be employed to identify the most important effects. In this paper, we introduce two straightforward strategies for effect pruning, as follows.

**Sparse Linear Models.** A simple approach for effect pruning is to fit a surrogate sparse linear model to identify and remove trivial effects. In this paper, we choose Lasso for regression tasks and L1 regularized logistic regression for classification tasks. The surrogate model can capture the overall relationships between the effects and their impact on the response. It can also identify and flag effects that contribute minimally or have a high correlation with other effects. These effects can be automatically pruned from the model, enhancing its interpretability by focusing on the most relevant and independent effects.

This pruning strategy shares a similar idea with the RuleFit algorithm (Friedman & Popescu, 2008). Both of them try to pursue a parsimonious representation of tree ensemble models by sparse linear modeling. The main difference lies in that RuleFit selects the most important decision rules, while ours performs pruning on the effects functions decomposed by functional ANOVA. Both of these methods are complementary and can also be combined. For instance, one may initially apply pruning on the decision rules level and subsequently represent the selected rules using functional ANOVA. However, such a combination is beyond the scope of this paper.

**Forward or Backward Effect Selection.** Another powerful strategy is called forward and backward selection with early dropping (FBEDk; Borboudakis & Tsamardinos, 2019). It consists of $k$ forward

---

**Algorithm 1** Pruning Functional ANOVA Effects

---

**Require:** Data $\{\boldsymbol{x}, y\}$, initial effects $\mathcal{F} = \{f_S\}$, performance gain threshold $\tau$, forward rounds $k$

1: **Step 1: Sparse Linear Screening**
2: Fit a sparse linear model between all $f_S(\mathbf{x}_S)$ and $y$, and let $\tilde{\mathcal{S}}$ be the currently selected effects.
3: **Step 2: Forward Selection with Early Dropping**
4: Let $\mathcal{C} \leftarrow \mathcal{F} \setminus \tilde{\mathcal{S}}$ be the candidate effects.
5: **for** iteration $r = 1$ to $k$ **do**
6:     **while** $\mathcal{C} \neq \emptyset$ **do**
7:         For each $f_S \in \mathcal{C}$, compute the performance gain of adding $f_S$ conditional on $\tilde{\mathcal{S}}$.
8:         Add the best $f_S$ to $\tilde{\mathcal{S}}$ if its performance gain $\geq \tau$.
9:         Remove all $f_S$ with performance gain $< \tau$ from $\mathcal{C}$.
10:     **end while**
11: **end for**
12: **Step 3: Backward Elimination**
13: **for** each $f_S \in \tilde{\mathcal{S}}$ **do**
14:     Compute performance gain of $f_S$ conditional on the rest selected effects.
15:     **if** gain $< \tau$ **then**
16:         Remove $f_S$ from $\tilde{\mathcal{S}}$.
17:     **end if**
18: **end for**
19: **Step 4: Effects Adjustment**
20: Fit an unconstrained linear or logistic model between $\tilde{\mathcal{S}}$ and $y$ to reduce bias.
21: **return** Pruned effect functions $\tilde{\mathcal{S}}$.

---

selection rounds and one backward elimination. The first forward selection round starts from a null model or a pre-defined effect set and iteratively adds effects that contribute significantly to the model's performance. The performance gain threshold $\tau$ controls whether an effect can be selected or dropped. In this paper, we use the R2 score for regression and the AUC for classification. Both of them range from 0 to 1, and we can empirically adjust the threshold from 1e-5 to 1e-3, to make sure selected effects do contribute to the model.

As $k > 1$, we would perform multiple rounds of forward selection, and each one starts from the selected effects of previous rounds. Due to the existence of the performance gain threshold, the length of the candidate effects list would become smaller and smaller within each round. Multiple forward rounds are used, as it is possible that one effect is not important in the first forward round but will become significant as conditioning on some other effects. Typically, $k = 2$ or $k = 3$ iterations are sufficient for the algorithm to converge and find a stable set of features. This is because the later iterations re-evaluate features that were dropped early in the first round but might be useful in combination with the newly selected features. Throughout this paper, we set $k = 2$.

Finally, as all the forward selection rounds are complete, we do a backward elimination round, starting from the least significant effects. This is testing whether the performance gain of each selected effect (conditioning on the rest selected effects) is greater than the threshold. Effects that fail this test are considered trivial and then removed from the model. As the effects are selected, we refit a generalized linear model between the selected effects and the target variable. The scale of each effect will be changed, while its shape will not. Note that the refitting step may make the model achieve better predictive performance.

**A Hybrid Approach.** In this paper, we use a hybrid approach that combines the above two strategies. First of all, we fit a sparse linear model to roughly select the important effects. Then, we treat the selected effects as initialization and use the FBEDk algorithm to fine-tune the results. It will assess the marginal contribution of each effect, and the ones with contributions greater or less than a pre-defined threshold will be added or deleted accordingly. Finally, given the selected effects, we refit a linear model without sparsity constraints to adjust the coefficients. See 1 for the pseudo codes of this hybrid approach.

Table 2: Predictive performance comparison.

| Task | Dataset | $n$ | $p$ | pyGAM | NAM | XGB-1 | EBM | GAMI-Net | XGB-2 | XGB-3 | XGB-5 |
|---|---|---|---|---|---|---|---|---|---|---|---|
| **REG** (RMSE) | friedman | 2000 | 10 | 1.359 ± 0.042 | 1.420 ± 0.042 | 1.423 ± 0.032 | 0.603 ± 0.043 | **0.149 ± 0.008** | 0.509 ± 0.030 | 0.571 ± 0.036 | 0.693 ± 0.048 |
| | bikesharing | 17379 | 8 | 0.662 ± 0.008 | 0.683 ± 0.009 | 0.662 ± 0.008 | 0.417 ± 0.010 | 0.439 ± 0.014 | 0.413 ± 0.008 | 0.401 ± 0.010 | **0.395 ± 0.008** |
| | wine quality | 1599 | 11 | 0.625 ± 0.016 | 0.621 ± 0.019 | 0.619 ± 0.022 | 0.602 ± 0.024 | 0.630 ± 0.021 | 0.609 ± 0.027 | 0.597 ± 0.027 | **0.583 ± 0.029** |
| | boston | 506 | 13 | 3.759 ± 0.615 | 4.122 ± 0.822 | 3.820 ± 0.781 | 3.757 ± 0.654 | 3.771 ± 0.634 | 3.237 ± 0.697 | **3.147 ± 0.742** | 3.381 ± 0.636 |
| | concrete | 1030 | 8 | 5.358 ± 0.791 | 6.973 ± 0.287 | 5.041 ± 0.392 | 4.156 ± 0.458 | 5.258 ± 0.324 | 4.331 ± 0.578 | 4.273 ± 0.537 | **4.088 ± 0.429** |
| | energy | 768 | 9 | 0.866 ± 0.284 | 4.880 ± 0.675 | 0.938 ± 0.067 | **0.534 ± 0.043** | 1.103 ± 0.344 | 0.551 ± 0.075 | 0.597 ± 0.106 | 0.670 ± 0.165 |
| | abalone | 4177 | 8 | 2.162 ± 0.085 | 2.227 ± 0.081 | 2.225 ± 0.068 | 2.232 ± 0.060 | **2.158 ± 0.108** | 2.184 ± 0.068 | 2.174 ± 0.061 | 2.202 ± 0.056 |
| **CLS** (AUC) | taiwancredit | 30000 | 18 | 0.773 ± 0.007 | 0.766 ± 0.008 | 0.772 ± 0.007 | 0.774 ± 0.007 | 0.770 ± 0.007 | 0.774 ± 0.006 | 0.774 ± 0.008 | **0.775 ± 0.006** |
| | creditsimu | 20000 | 7 | 0.740 ± 0.022 | 0.743 ± 0.006 | 0.745 ± 0.009 | 0.753 ± 0.007 | 0.753 ± 0.008 | **0.754 ± 0.008** | 0.752 ± 0.006 | 0.753 ± 0.008 |
| | adult | 48842 | 14 | 0.912 ± 0.003 | 0.907 ± 0.003 | 0.912 ± 0.003 | 0.914 ± 0.003 | 0.910 ± 0.003 | 0.913 ± 0.003 | **0.914 ± 0.003** | 0.914 ± 0.003 |
| | bank | 45211 | 16 | 0.916 ± 0.004 | 0.901 ± 0.002 | 0.916 ± 0.004 | 0.930 ± 0.003 | 0.911 ± 0.005 | 0.932 ± 0.003 | 0.935 ± 0.002 | **0.936 ± 0.003** |
| | compas | 5278 | 13 | 0.731 ± 0.014 | 0.734 ± 0.015 | 0.734 ± 0.014 | **0.735 ± 0.013** | 0.734 ± 0.013 | 0.733 ± 0.013 | 0.733 ± 0.014 | 0.731 ± 0.013 |
| | magic | 19020 | 10 | 0.908 ± 0.008 | 0.903 ± 0.006 | 0.908 ± 0.007 | 0.936 ± 0.004 | 0.920 ± 0.009 | 0.939 ± 0.004 | 0.940 ± 0.005 | **0.940 ± 0.003** |
| | titanic | 2201 | 3 | 0.744 ± 0.028 | 0.738 ± 0.024 | 0.744 ± 0.028 | 0.757 ± 0.029 | 0.744 ± 0.026 | 0.757 ± 0.030 | **0.757 ± 0.029** | 0.757 ± 0.029 |

## 5 NUMERICAL RESULTS

Among the tree ensemble models, we choose the XGB model implemented by the *xgboost* package throughout the experiments. As maximum depth is the most important hyperparameter, we abbreviate XGB with max depth 1 as XGB-1, and XGB with max depth 2 as XGB-2, etc. The detailed experiment setup can be found in Appendix C.

### 5.1 PREDICTIVE PERFORMANCE COMPARISON

To comprehensively evaluate the predictive accuracy of competing models, we conduct experiments on a diverse collection of publicly available benchmark datasets spanning both classification and regression tasks. For regression, we evaluate performance on the friedman simulation dataset and six widely used real-world datasets, including bike sharing, wine quality, boston, concrete, energy, and abalone. The classification suite includes taiwancredit, creditsimu, adult, bank, compas, magic, and titanic, covering a broad range of sample sizes. These datasets represent typical tabular learning scenarios in housing prices, credit scoring, socio-economic prediction, healthcare risk assessment, etc.

The results in Table 2 show that across both regression and classification tasks. The best results are highlighted in bold, while statistically close results are underlined. For regression datasets, XGB-2 often ranks among the top methods, with RMSE only slightly higher than the best values in datasets like Boston, Concrete, Energy, and Abalone. Similarly, for classification datasets, XGB-2 reaches AUC values comparable to or just below the top-performing models. Importantly, while achieving near state-of-the-art predictive performance, XGB-2 remains highly interpretable, striking a favorable balance between accuracy and model transparency. This makes it a strong candidate for applications where interpretability is critical without substantially sacrificing performance.

### 5.2 CASE STUDY: FRIEDMAN DATASET

The Friedman data is generated using the following simulation function as described in (Friedman, 1991; Breiman, 1996).

$$y(\boldsymbol{x}) = 10 \sin(\pi x_1 x_2) + 20 (x_3 - 0.5)^2 + 10 x_4 + 5 x_5 + \varepsilon, \tag{10}$$

where $\varepsilon \sim N\left(0, \sigma^2\right)$. The covariates are uniformly distributed between 0 and 1. In this experiment, we simulate data with $n = 2000$ and $\sigma = 0.1$. In addition, we introduce another 5 noise features ($x_6$ to $x_{10}$) when generating the data. We first fit an XGB-2 model using training data, and then transform it into a functional ANOVA representation with 10 main effects and 45 pairwise interactions. After that, we use Lasso with different regularization strengths to reveal the relationship between predictive performance and the number of selected effects in Figure 2. The x-axis is the regularization strength; the bar chart (on the left y-axis) shows the number of selected effects, and the line plot (on the right y-axis) displays the 5-fold cross-validation R-squared (R2) score. From the results, it can be observed that R2 reaches its maximum when the regularization is small (from 0.001 to 0.009). As the regularization strength increases to 0.078, the selected effects suddenly shrink to 5

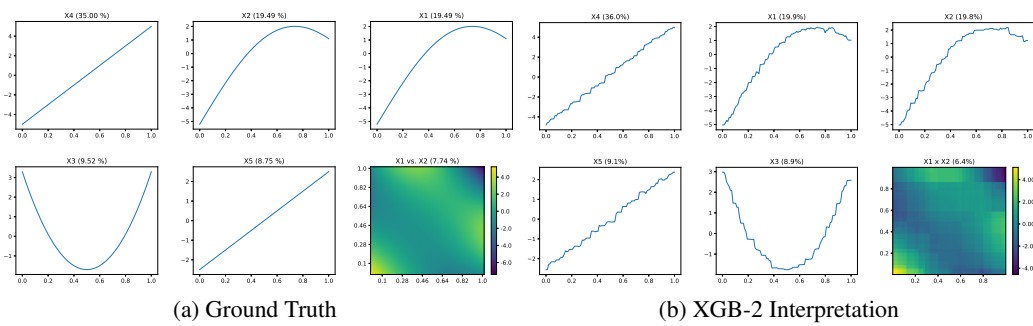

(a) Ground Truth                    (b) XGB-2 Interpretation

Figure 3: The fitted results of XGB-2 vs. the ground truth of the Friedman dataset.

main effects and 1 pairwise interaction, while the R2 score does not change too much. This means that the rest 5 main effects and 44 pairwise interactions are trivial and can be pruned.

Inspired by this, we do post-hoc effect pruning by fitting a Lasso with a regularization strength of 0.078, and then fine-tune the selected effects by the FBEDk algorithm. It is worth mentioning that after effect pruning, the test set RMSE gets improved to around 0.425. This means that removing the trivial effects can not only enhance model interpretability but also mitigate overfitting.

Figure 3 displays the obtained main effects and pairwise interactions after effect pruning, together with the ground truth functions. For each effect plot, we show the corresponding effect importance in the title. Overall, the effects fitted by XGB-2 are close to the actual functions, and the difference is due to the inherent model form of tree ensemble models, i.e., the piece-wise constant model fits. More details about global and local feature / effect importance can be found in Appendix D.

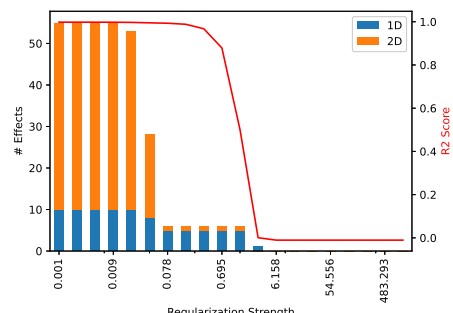

Figure 2: The number of selected effects (y-axis) and 5-fold cross-validation performance under different regularization strengths of Lasso for the Friedman dataset.

### 5.3 CASE STUDY: BIKE SHARING DATASET

This dataset [1] records the hourly count of rental bikes in the Capital bikeshare system from 2011 to 2012. It has 17389 samples, each data record captures the weather and seasonal conditions within an hour, and the task is to predict the total rental bikes, including both casual and registered. For modeling purposes, we remove some of the highly redundant variables. The selected predictors include season, hr (hour of a day), holiday (whether the day is a holiday or not), weekday (day of the week), weathersit (weather conditions), atemp (normalized feeling temperature), hum (normalized humidity), and windspeed (normalized wind speed). As the response is counting data, we process it using a log transformation.

We first fit an XGB-3 model, and then analyze the relationship between the number of effects and the predictive performance using the Lasso regularization path plot, as shown in Figure 4. Based on this plot, we conduct post-hoc effect pruning with Lasso (regularization strength equals 0.005) and FBEDk (for fine-tuning). Finally, the pruned model has 8 main effects, 37 pairwise interactions, and 46 3-way interactions. The pruned model has a test set RMSE of around 0.408, which is close to that of the raw XGB-3 model (0.406, with the same random seed).

---

[1] https://archive.ics.uci.edu/dataset/275/bike+sharing+dataset

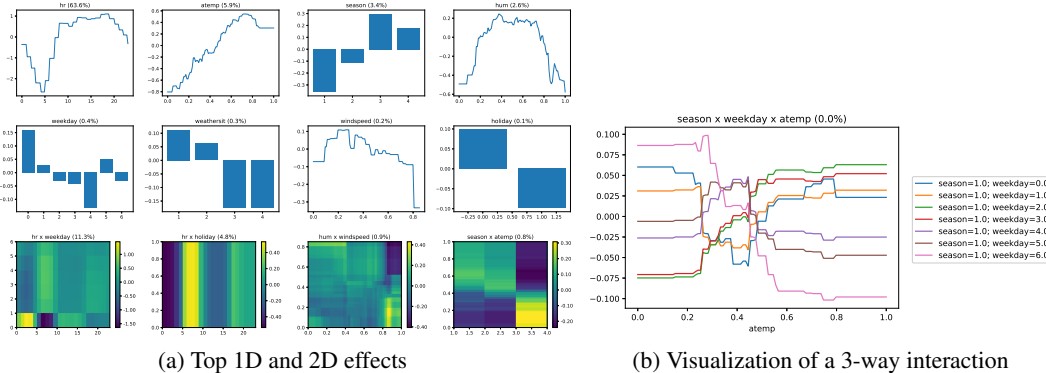

(a) Top 1D and 2D effects          (b) Visualization of a 3-way interaction

Figure 5: The fitted results of XGB-2 vs. the ground truth of the Friedman dataset.

The decomposed effects of the final model is displayed in Figure 5, which shows the most important main effects and pairwise interactions after pruning. For 3-way interactions, we can use the sliced 1D plot to reveal the patterns. For example, the most important 3-way interaction is season, weekday, and atemp. It's interesting to note that even the most important 3-way interaction has an effect importance close to zero, which indicates that high-way interactions are in general less important than main effects and 2-way interactions. Conditioning on different value combinations of season and weekday, we draw this interaction value against atamp in Figure 5. This plot only uses season=1.0, and the other values of season can also be drawn in other plots. It reveals that atemp has an increasing trend to the target as season=1.0 and weekday is 2.0 or 3.0 (Tuesday or Wednesday); however, a decreasing trend is observed as weekday is 6.0 (Saturday).

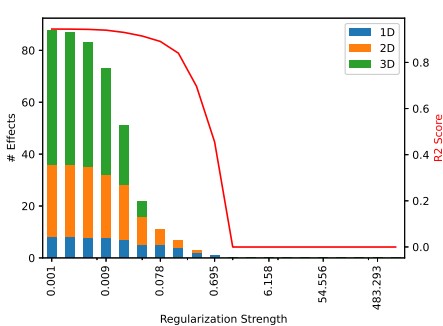

Figure 4: The number of selected effects (y-axis) and 5-fold cross-validation performance under different regularization strengths of Lasso for the BikeSharing dataset.

## 6   CONCLUSION

This paper proposes an interpretation algorithm to open the black box of tree ensemble models. Based on the functional ANOVA framework, a fitted tree ensemble model can be equivalently converted into the generalized additive model with interactions. Each of the decomposed main effects and pairwise interactions can be easily interpreted and visualized. Multi-way interactions are more difficult to interpret; however, we empirically show that they are less important and sometimes can be pruned without sacrificing too much predictive performance.

A notable limitation of the proposed approach is its difficulty in capturing interactions beyond third-order. While this restriction is sufficient for many tabular datasets, it constrains the model's applicability in domains where higher-order or more complex feature interactions play a critical role, such as in computer vision or natural language understanding. This limitation arises from the combinatorial growth of interaction terms and the corresponding challenges in estimation and interpretability. Future work could address this by exploring strategies to efficiently approximate or selectively model higher-order interactions.

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

## A EXTENDED RELATED WORKS

The techniques in interpretable machine learning can be roughly classified into post-hoc explanation tools and inherently interpretable models. The former aims at explaining an arbitrary model, and it can be further divided into global and local explanations. Examples of global explanation include partial dependence plot (PDP; Friedman, 2001) and accumulative local effects (ALE; Apley & Zhu, 2020), where both of them are used to reveal the relationship between one or two features and the model prediction. In contrast, local explanation methods like the local interpretable model-agnostic explanation (LIME; Ribeiro et al., 2016) and Shapley additive explanations (SHAP; Lundberg & Lee, 2017; Lundberg et al., 2020) decompose the prediction outcome of an individual sample into the contributions of each feature. The primary drawback of post-hoc explanation tools is that the interpretation results are mere approximations, which may deviate from the original model and be incorrect or unfaithful (Rudin, 2019). This can pose significant risks, especially in sensitive domains such as healthcare and finance.

The second category aims at developing inherently interpretable models. This is in contrast to black-box models (e.g., neural networks), in which the decision-making process is too complicated to interpret. In practice, much of the complexity is unnecessary and may lead to overfitting. The key idea of inherently interpretable models is to regularize or constrain complex models to be interpretable, without sacrificing predictive performance. Some principles of designing interpretable models include additivity, sparsity, smoothness, etc (Sudjianto & Zhang, 2021).

For instance, an explainable boosting machine (EBM; Lou et al., 2013) is a generalized additive model (GAM) with functional pairwise interactions. It fits the main effects and interactions sequentially using shallow tree ensemble models. The generalized additive model with structured pairwise interactions network (GAMI-Net; Yang et al., 2021) is an alternative to EBM, but uses modularized neural networks to estimate the main effects and pairwise interactions. The GAMI-Lin-T (Hu et al., 2023) model is another recently proposed interpretable model under the functional ANOVA framework. It also uses the boosting algorithm, and the base learners are trees with linear functions in leaves.

More recent work has explored combining the flexibility of deep learning with the transparency of GAMs. Neural Additive Models (NAMs; Agarwal et al., 2021) extend classical GAMs by training a separate neural subnetwork per feature and summing their outputs, offering deep learning expressivity while preserving clear per-feature shape functions. NODE-GAM and NODE-GA$^2$M (Chang et al., 2021) further improve scalability by introducing differentiable architectures for GAMs and GA$^2$Ms that can handle large datasets and benefit from modern optimization techniques. Neural Basis Models (NBM; Radenovic et al., 2022) address the parameter inefficiency of NAMs by learning a small shared set of basis functions across features. Similarly, Scalable Polynomial Additive Models (SPAM; Dubey et al., 2022) employ tensor decompositions to compactly model higher-order interactions. Sparse Interaction Additive Networks (SIAN; Enouen & Liu, 2022) focus on detecting and selecting only a small subset of important interactions, balancing interpretability and predictive power. Extensions such as NAM-LSS (Thielmann et al., 2024) incorporate probabilistic modeling by predicting not only the mean but also other distributional parameters, while GAMformer (Mueller et al., 2024) leverages transformers to perform amortized inference of GAM components. These developments highlight a growing trend toward models that remain inherently interpretable while offering scalability and accuracy comparable to complex black-box models.

The proposed interpretation algorithm combines elements from both categories mentioned above, serving as a post-hoc tool specifically for interpreting tree ensemble models. Notably, it endows tree ensemble models with inherent interpretability, ensuring the derived interpretations are precise without any approximation. Additionally, a recently introduced effect purification algorithm (Lengerich et al., 2020) is incorporated to tackle the identifiability problem between main effects and their corresponding interaction effects under the functional ANOVA framework. This paper leverages this purification algorithm to convert tree ensemble models into a functional ANOVA-based representation.

In the literature, there exist some attempts to interpret shallow tree ensemble models. For example, the decision stump boosting (Oliver & Hand, 1994; Denison, 2001) uses decision trees with only one split as base learners, and the resulting model can be represented as a generalized additive model. The EBM models share the same model form as tree ensemble models, as the maximum tree depth

is 2. Both of them are composed of main effects and pairwise interactions, and the effect functions are piecewise constant. The main difference is in the model fitting method. In EBM, the main effects are fitted first in a round-robin fashion, and followed by the pairwise interactions. In contrast, tree ensemble models fit all effects greedily without any predefined order, and, therefore, tend to have better predictive performance.

Our work is also highly related to Molnar et al. (2019), where the authors introduce a post-hoc tool to measure model complexity, by leveraging ALE and functional ANOVA to approximate main effects and interaction strength. In Schneider et al. (2023), post-hoc interpretability metrics like feature sparsity, interaction sparsity, and monotonicity are proposed, and a multi-objective optimization framework is designed to search for a better trade-off of model interpretability and predictive performance. Compared to these 2 methods, our work is a complement and an alternative that provides exact, faithful interpretations for tree ensemble models by construction. a) In the model decomposition stage: By constraining the model to be a shallow tree ensemble, we can perform an exact functional ANOVA decomposition. We identify main effects and interactions exactly with a precise functional form, without approximating them with ALE or PDPs. Our explanation is guaranteed to match the model's output exactly, for every prediction; b) In the pruning stage, we provide the option to further enhance the fitted model, towards a more interpretable / robust model. Beyond reporting a sparsity score (like Schneider et al.), our enhanced model can still be directly visualized and interpreted for every main effect and interaction.

## B  INTERPRETABILITY-ORIENTED HYPERPARAMETERS

This appendix provides full descriptions and practical considerations for the hyperparameters listed in Table 1, which can be adjusted to enhance the interpretability of tree ensemble models.

### B.1  MAXIMUM TREE DEPTH

In the full functional ANOVA representation, the total number of effects is $2^p - 1$. This number would become extremely large with the increase of $p$. If all the effects are active or non-zero, then the resulting model can be very complicated and hard to interpret. Fortunately, in tree ensemble models, we can easily control the highest interaction order by maximum tree depth, which is a commonly used hyperparameter. For example, as the maximum tree depth is 1, then all the interaction effects are zero, and the model reduces to a generalized additive model (GAM) with at most $p$ main effects; as the tree depth is 2, then the model would only have main effects and pairwise interactions, which has same model form as the explainable boosting machine (EBM). In this case, the total number of effects is less than or equal to $p(p+1)/2$. As not all the features are used as split variables, the number of active effects is usually smaller than the number of possible effects.

With a maximum tree depth of 3, we can still interpret the interactions involving 3 features using 3D heatmaps or sliced 1D plots. For example, we can examine a 3-way interaction by visualizing one or two features while keeping the rest one or two features fixed at certain values. However, as the tree depth increases, the model's complexity grows exponentially, making it more challenging to interpret deep tree ensemble models.

In addition, deep tree ensembles are hard to interpret, also from an algorithmic perspective. If given adequate computing resources, the purification algorithm can be applied to arbitrary interaction effects. However, the tensors representing high-order interactions tend to become excessively large, making them difficult to process. In practical scenarios, purifying interactions involving 4 or more features becomes challenging, and sometimes even impossible. Hence, to maintain feasibility, our interpretation in this paper is restricted to depth-3 tree ensemble models.

In practice, well-configured shallow tree ensemble models are often sufficient to achieve good predictive performance. It's worth noting that when we limit the maximum depth of base tree learners, it is recommended to increase the number of estimators (boosting rounds). This is because shallow trees in nature have much lower expressive power compared to deeper ones. For instance, a depth 2 tree ensemble model with 100 estimators would have at most 400 leaf nodes, while a similar depth 5 model would have at most 3200 leaf nodes. Therefore, to compensate for the reduction in tree depth, we may need to increase the number of estimators.

## B.2 MONOTONICITY

In many real-world applications, enforcing feature monotonicity in a model is highly desirable for interpretation purposes. In a credit scoring model, it is expected that applicants' credit scores increase monotonically with their income. However, in practice, this assumption can be easily violated due to noisy data, rendering the model difficult to interpret and diminishing people's trust in its predictions. In tree ensemble models, monotonicity constraints can be imposed in fitting each tree. For instance, to make a feature monotonic increasing, we can prohibit candidate splits of that feature where the resulting left child node value is greater than that of the right one.

The monotonicity constraint can be specified by leveraging domain knowledge before model training. It can significantly enhance the interpretability and trustworthiness of the model. On the other hand, the EBM model, as a counterpart benchmark, lacks inherent monotonicity constraints, and adjustments can only be made post-training. Such post-hoc adjustments may introduce bias and potentially decrease the overall performance of the model.

## B.3 MAXIMUM NUMBER OF BINS

This parameter is preliminarily employed to reduce the search space of split points. Instead of considering all possible unique feature values as candidate split points, it selects a predetermined number of quantiles for each feature as candidates. From the perspective of model interpretability, restricting the number of bins can also prevent unnecessary discontinuities and make the estimated effects more easily comprehensible. Therefore, this hyperparameter is very useful in practical applications.

## B.4 INTERACTION CONSTRAINT

Certain tree ensemble learning frameworks provide an API that allows for the restriction of candidate feature interactions. By using this option, interactions outside of a predefined list of interactions can be prohibited. For example, if we specify the allowed interactions as $(x_1, x_2)$ and $(x_2, x_3)$, the resulting fitted model would not include interactions such as $(x_1, x_3)$. This feature is useful when we possess prior or domain knowledge about the data being modeled, or when we just aim to reduce the complexity of the model.

It is important to note that by applying the feature interaction constraint, the maximum tree depth parameter can be relaxed and set to a larger value without increasing the highest order of interactions. For example, if our goal is to include only main effects and pairwise interactions, we can set the maximum tree depth to a value greater than 2, while constraining the interaction list to encompass all possible pairwise interactions. This approach provides flexibility in hyperparameter tuning, allowing us to vary the depth of the trees while still capturing the desired level of interactions. By using this trick, we can strike a balance between model complexity and interpretability, tailoring the model to our specific requirements.

## B.5 MISCELLANEOUS

There are several other hyperparameters that can be utilized to enhance the interpretability of tree ensemble models. Here, we outline some of the commonly employed ones:

**L1 / L2 Regularization.** Similar to the regularization techniques used in linear models, the application of L1 or L2 regularization can help penalize large values in leaf nodes. By increasing the regularization strength, insignificant leaf nodes can be eliminated, effectively reducing their impact to zero. Consequently, the functional ANOVA representation will also become sparser, making it easier to interpret.

**Early Stopping Conditions.** In addition to the aforementioned criteria, certain early stopping conditions can also be considered as interpretability constraints. These include hyperparameters such as the minimum number of samples per leaf, the minimum loss reduction required for splits, the number of rounds for early stopping, and so on.

## C  EXPERIMENT SETUP

For comparison, the spline-based GAM, Neural additive model (NAM), EBM, GAMI-Net, and XGB-5 are included as benchmarks. The spline-based GAM is implemented by the *pyGAM* Python package (Servén & Brummitt, 2018). The NAM is implemented by the *nam* Python package (Kayid et al., 2020). The EBM model is implemented in the Python package *interpret* (Nori et al., 2019). The GAMI-Net model is based on the implementation in the *PiML* Python package [2]. Moreover, the proposed tree ensemble model interpretation algorithm is also integrated into the *PiML* package.

We randomly split each dataset into training (80%) and test (20%) sets. For hyperparameter tuning and monitoring the early stopping criteria, 20% of the training samples are used for validation purposes. For each XGBoost model, we tune the number of estimators (50 to 3000), learning rate (0.01 to 1), L1 regularization (0.001 to 1000), L2 regularization (0.001 to 1000), and maximum number of bins (2 to 200). In pyGAM, we tune the spline order (0 to 3), number of splines (10 to 50), and smoothing penalty (0.001 to 1000). In EBM, we tune the number of interactions (0 to 100) and the learning rate (0.01 to 1). For each model, we tune the hyperparameters using the random search strategy (Bergstra & Bengio, 2012), and the number of trials is set to 30. Specifically, we randomly generate 30 hyperparameter configurations for each model within the search space; the one that achieves the best validation performance is selected, and then we refit the model using all the training data. For speed consideration, most hyperparameters in NAM are set to default, and we empirically set the activation function to ReLU, and the maximum epoch to 200. GAMI-Net is configured and trained using the default settings, with the number of interactions fixed to 10.

The predictive performance is measured by the root-mean-square error (RMSE) for regression tasks and the area under the ROC curve (AUC) for binary classification tasks. All the experiments are repeated 10 times.

## D  EXTENDED CASE STUDIES

### D.1  MORE RESULTS OF FRIEDMAN DATASET

Figure 6a and Figure 6b show the effect and feature importance defined in Section 4. The 5 main effects $X_1, X_2, \cdots, X_5$ are most important to the model prediction, followed by the interaction $X_1 \times X_2$. The feature importance further aggregates the contribution of interactions to each feature. It turns out that $X_4$ is the most important, $X_2, X_1$ are less important, and $X_5, X_3$ are of the least importance.

Given a specific sample, the local explanation tries to explain how the model generates its prediction. The prediction can be additively decomposed into effect contributions and feature contributions, see a demo in Figure 6c and Figure 6d. The left axis is the effect / feature names, the right axis shows the feature values of the given sample, and the bar charts represent the contributions of each effect / feature to the prediction. In the title, we also give the predicted value and the actual response.

### D.2  CASE STUDY: CREDITSIMU DATASET

This example is a credit decision dataset [3] with synthetic features of applicants, including Mortgage (mortgage size), Balance (average credit card balance), Amount Past Due (minimum required payment that was not applied to the account as of the last payment due date), Delinquency status (0: current, 1: less than 30 days delinquent, 2: 30-60 days delinquent, 3: 60-90 days, etc), Credit Inquiry (number of credit inquiries), Open Trade (number of open credit accounts), and Utilization (credit utilization ratio). This data is provided in the *PiML* package, and the response feature is binary, indicating whether the application is approved or not.

According to our domain knowledge, it is expected that some of the features are monotonic with respect to the credit card approval rate. In addition to the raw XGB-2 model, we fit another XGB-2 model with enhanced interpretability constraints. Specifically, we constrain the Mortgage to be

---

[2]`https://github.com/SelfExplainML/PiML-Toolbox/`
[3]`https://github.com/SelfExplainML/PiML-Toolbox/blob/main/datasets/`
`SimuCredit.csv`

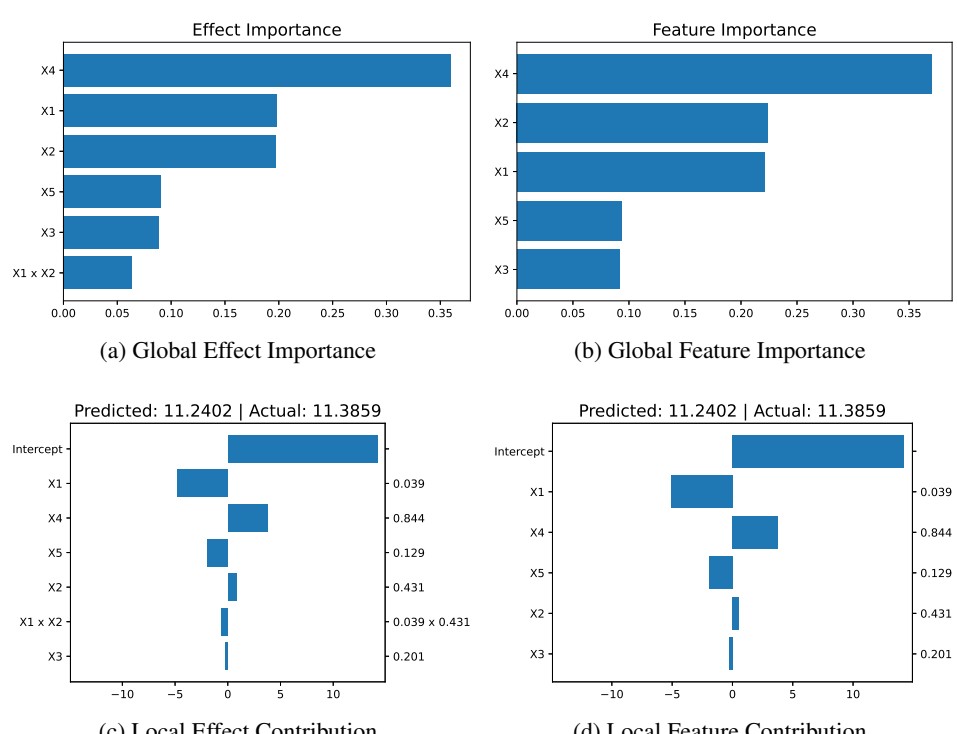

Figure 6: The effect and feature importance of the Friedman dataset.

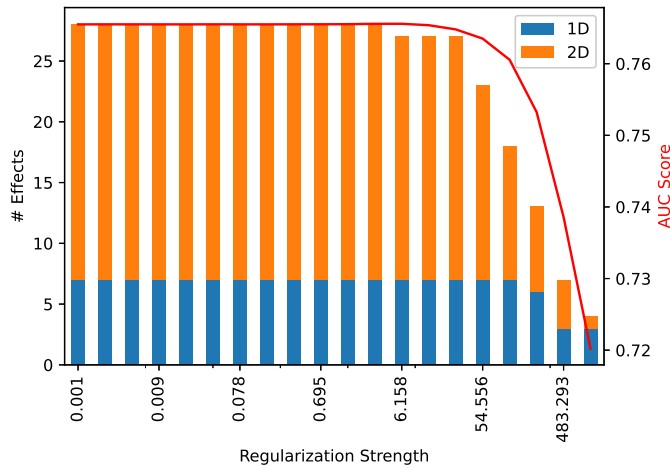

Figure 7: The number of selected effects (y-axis) and 5-fold cross-validation performance under different regularization strengths of Lasso for the CreditSimu dataset.

monotonically increasing and Utilization to be monotonically decreasing. We also limit the maximum number of bins to 20 to avoid unnecessary jumps in fitted shape functions. The constrained XGB-2 model still achieves a test AUC score of around 0.754. This means that the interpretability constraint does not come with any sacrifice in predictive performance.

Finally, we also show the Lasso regulation path of the constrained XGB-2 upon functional ANOVA decomposition in Figure 7. According to the trade-off between model sparsity and AUC score, we prune the constrained XGB-2 via L1-regularization logistic regression (with regularization strength equal to 10) and FBEDk (for fine-tuning). The pruned model has a test AUC score of around

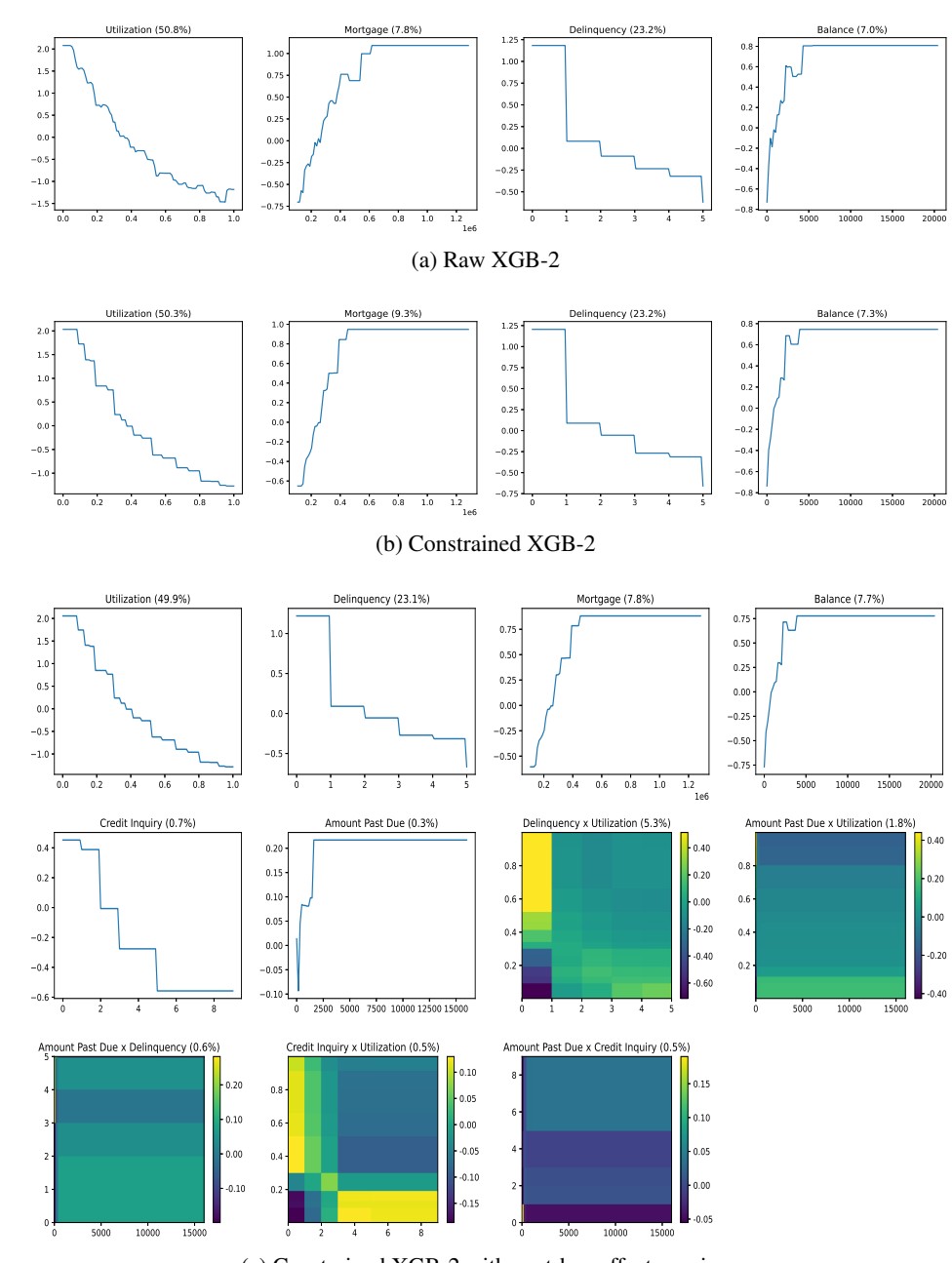

(a) Raw XGB-2

(b) Constrained XGB-2

(c) Constrained XGB-2 with post-hoc effect pruning

Figure 8: The visualization comparison of different versions of XGB-2 on the CreditSimu dataset.

0.753 but only includes 7 main effects and 15 pairwise interactions. In Figure 8, we show the extracted main effects and pairwise interactions of the pruned XGB-2 model. Note that this is a binary classification task, and hence the y-axis is the log odds ratio. For comparison purposes, we also display the effects of the raw XGB-2 and constrained XGB-2. For simplification, we only show the effects of Mortgage, Utilization, Delinquency, and Balance. It can be found that these constraints make the shape functions of main effects less jumpy and easier to interpret.

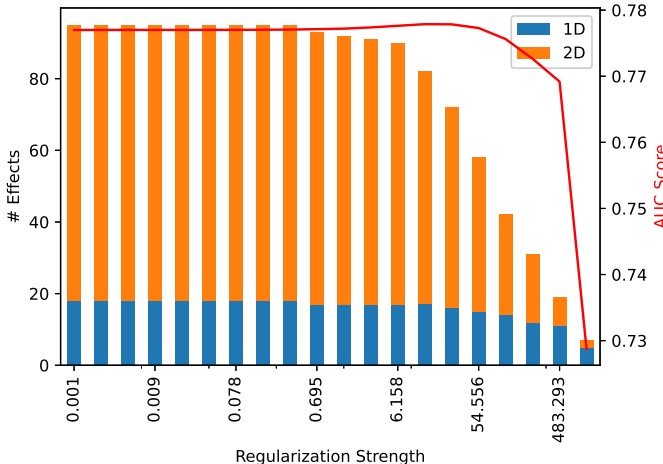

Figure 9: The number of selected effects (y-axis) and 5-fold cross-validation performance under different regularization strengths of Lasso for the TaiwanCredit dataset.

### D.3 CASE STUDY: TAIWANCREDIT DATASET

TaiwanCredit data [4] is obtained from the UCI repository, which consists of 30,000 credit card clients in Taiwan from 200504 to 200509. In this experiment, we only use the 18 payment features as predictors, including Pay_1 to 6 (past payment delay status), BILL_AMT1 to 6 (amount of bill statement), and PAY_AMT1 to 6 (amount of previous payment). Note that Pay_1 is renamed from Pay_0 in the original data. The target variable is default payment, with 1 indicating default payment.

Similar to the previous analysis, we first add monotonically increasing constraints for Pay_1 to 6 (history of past payment) and BILL_AMT1 to 6; while PAY_AMT1 to 6 are enforced to be monotonically decreasing. In addition, the maximum number of bins is set to 20. With these interpretability constraints, the constrained XGB-2 model has a test set AUC score of around 0.772, which is slightly lower than that of the raw XGB-2 (around 0.774). The regularization path of the constrained XGB-2 model is drawn in Figure 9. Using this plot, we set the regularization strength of L1-regularized logistic regression to 15 and fine-tune the selected effects by FBEDk. The final pruned model has 15 main effects and 24 pairwise interactions, which is much smaller than the non-pruned model (18 main effects and 97 pairwise interactions). Meanwhile, the pruned model also has a test set AUC score of around 0.772.

Figure 10 shows the effects visualization of the 3 versions of XGB-2 models. Clearly, with enhanced interpretability constraints, the effect functions look much more reasonable compared to the unconstrained ones. This further verified our belief that properly imposed constraints can make machine learning models more interpretable.

---

[4]https://archive.ics.uci.edu/dataset/350/default+of+credit+card+clients

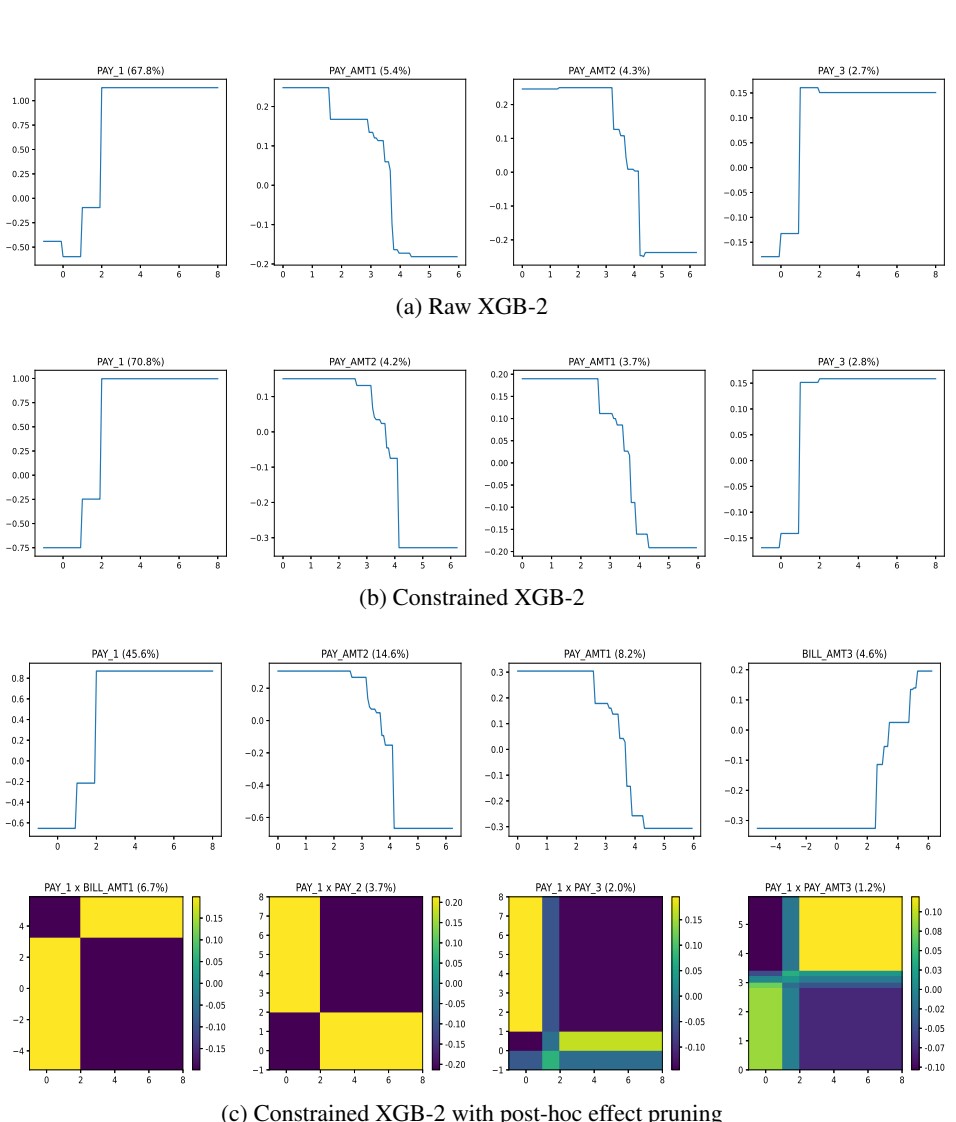

(a) Raw XGB-2

(b) Constrained XGB-2

(c) Constrained XGB-2 with post-hoc effect pruning

Figure 10: The visualization comparison of different versions of XGB-2 on the TaiwanCredit dataset.

