# OpenReview forum: "Inherently Interpretable Tree Ensemble Learning"
_ICLR.cc/2026/Conference — Submitted to ICLR 2026_

### Official Review · Reviewer_qYwq · 2025-10-28

**Soundness:** 2
**Presentation:** 1
**Contribution:** 2
**Rating:** 2
**Confidence:** 4

**Summary:**

The paper shows that ensembles of shallow decision trees admit an ANOVA decomposition into multiple factors. The pipeline consists of: constrain training for interpretability, decompose and purify to obtain identifiable effects, and prune trivial terms. Experiments on a synthetic benchmark illustrate the accuracy-interpretability trade‑off.

**Strengths:**

S1. The focus on interpretability for tabular prediction is important and timely.

S2. The paper includes a clear complexity analysis of the purification step and discusses when higher‑order interactions become infeasible.

**Weaknesses:**

W1. The paper is not easy to read in places. The method section is verbose; for instance, Section 4.3 is entirely prose and would benefit from concise pseudocode or an algorithm box. The meaning of “#Effects” in Table 2 is not explained clearly enough for a reader to infer exactly how effects are counted or averaged.

W2. The abstract promises real‑data experiments, but the main text only presents a synthetic study; real‑data results are deferred to the appendix. If these results are central, they should be moved into the main paper.

W3. The baselines omit recent neural additive/interpretable models such as NAM and Gamformer, which would strengthen the empirical comparison.

**Questions:**

Q1. What are the hyperparameters of the proposed pipeline in practice, and how were they chosen in the experiments (search space, selection criteria, and validation protocol)?

Q2. On real data the ground‑truth structure is often unknown, so validating recovered effects as in Figure 4 is not straightforward. How do you propose to evaluate interpretability on real‑world datasets (for example, stability of discovered effects, agreement with domain constraints, or other quantitative proxies)?

---

### Official Review · Reviewer_Pvxn · 2025-10-30

**Soundness:** 3
**Presentation:** 4
**Contribution:** 2
**Rating:** 4
**Confidence:** 3

**Summary:**

This work proposes a framework for building inherently interpretable tree ensembles by restricting trees to very low depth. The intuition comes from viewing the model through its functional ANOVA decomposition: limiting tree depth directly restricts the order of interaction terms (e.g., depth-2 trees induce at most second-order interactions). The approach is conceptually simple: train tree ensembles with a fixed shallow depth so that the resulting model naturally has a low-order interaction structure, and apply a pruning step to remove redundant components. Experiments, primarily on the Friedman synthetic dataset (with additional results on CreditSimu in the appendix), show that shallow tree ensembles maintain competitive accuracy relative to EBMs (which disallow interactions) and GAMs, while offering an interpretable, low-interaction representation.

**Strengths:**

1. The approach of constraining tree ensembles during training to encourage inherent interpretability is interesting, particularly given the paper’s connection to the functional ANOVA perspective.

2. The observation that shallow tree ensembles correspond to low-order functional ANOVA components is practically useful and interesting, though not particularly surprising. It is also unclear how novel this insight is relative to prior work.

3. The paper is well written, clearly structured, and *very* easy to follow, making it highly accessible.

4. While the empirical findings suggest that low-order functional ANOVA tree ensembles can sometimes have comparable accuracy, a deeper evaluation and many more comparisons are needed (see Weaknesses).

**Weaknesses:**

1. A central limitation is the lack of sufficient experimental evaluation. Relying on a single synthetic benchmark in the main text and one additional dataset in the appendix does not provide enough evidence to support the claims. In particular, more comprehensive experiments are needed, especially on benchmarks known to require higher-order interactions.

2. The empirical study only compares against GAMs and does not include comparisons to high-order GAMs or EBMs, which are highly relevant baselines for this setting.

3. It is not fully clear from the theoretical discussion whether a tree ensemble of depth-$k$ is functionally equivalent to an EBM with up to $k$ interactions. If they are equivalent, the paper should address why depth-constrained trees are preferable to directly training an EBM with $k$-order interactions, both theoretically and empirically. If they are not equivalent, the paper should clearly articulate the differences and explain what interpretability or modeling advantages the proposed tree-based approach offers that high-interaction EBMs do not.

4. The observation that tabular datasets typically do not require high-order interactions is well-established in prior work. For example, [1] already demonstrated that EBMs with only low-order interactions (up to two) achieve performance competitive with full tree ensembles.

5. The method appears to struggle when moving beyond second- or third-order interactions, which limits its applicability primarily to tabular tasks rather than domains requiring richer interactions (e.g., vision). It would strengthen the work to discuss this limitation more explicitly and to outline possible directions for extending the approach to handle deeper interaction structure.

**Questions:**

1. If a tree ensemble is constrained to depth k, is it functionally equivalent to an EBM that includes interaction terms up to order k? If so, what benefit is gained by training a constrained-depth tree ensemble instead of directly using an EBM with interactions of the same order?

2. Was it previously known that depth-k tree ensembles induce a functional ANOVA decomposition of order k? This seems like a natural and expected property, so it is surprising that prior work does not appear to emphasize it.

3. The paper notes that moving from depth 2 to depth 3 leads to a combinatorial explosion that makes computing the functional ANOVA infeasible. A recent study [2] also showed that generating several types of explanations for tree ensembles already with depth 3 becomes NP-hard. How do these results connect? Is there a true phase transition between 2 and 3, or is the practical intractability simply due to the jump from $O(n^2)$ to $O(n^3)$ interactions, which makes it impossible to enumerate in practice?

4. Beyond limiting tree depth, what other structural constraints could be imposed on a tree ensemble to guarantee inherent interpretability?

[1] Intelligible Models for HealthCare: Predicting Pneumonia Risk and Hospital 30-day Readmission (Caruana et al., KDD 2015)

[2] What makes an Ensemble (Un) Interpretable? (Bassan et al., ICML 2025)

---

### Official Review · Reviewer_tGa8 · 2025-10-30

**Soundness:** 2
**Presentation:** 3
**Contribution:** 3
**Rating:** 4
**Confidence:** 3

**Summary:**

This work focuses on the topic of inherently interpretable models. Specifically, the authors propose a method of translating tree-ensemble models with a functional ANOVA-style decomposition into a sum of simple (piece-wise constant) components. They discuss various hyperparameters influencing the interpretability of tree ensembles, introduce the pipeline, and evaluate on a few datasets.

**Strengths:**

The paper reads quite well; it flows logically. The connection between tree ensembles and the simple additive components is quite interesting. The discussion of interpretability-relevant parameters and the analysis of the dependence of performance on the number of components used are insightful. The method seems to perform okay.

**Weaknesses:**

The Numerical Results are the paper's biggest weakness.
In contrast to the claim of "superior trade-off between interpretability and predictive power on synthetic and real-world datasets":
- The performance is not shown to be superior to the compared methods; they seem to (in some cases) show better performance with fewer components. Classification performance is also evaluated only with AUC.
- The real-world data results are missing from the main body.
- Statistical significance was not tested.
- The method comparison lacks non-GAM interpretable models (e.g., datasets) and some uninterpretable benchmark methods (e.g., standard XGBoost) to compare performance.
- Only the number of components is considered as a proxy for interpretability, while the interpretability of components themselves is not systematically interrogated, other than a few case studies. User study was not performed.
- There are also claims of practicality and high performance of the mehtod (L23). The evaluation in the main body does not discuss real datasets and the real datasets in appendix
The proposed technique would certainly benefit from a more extensive evaluation using multiple tree ensemble learners, multiple datasets of various sizes, comparing to other interpretable models, including the performance of uninterpretable base models, and ideally including a user study to compare interpretability.

On lines 206-210, I believe that the functions cannot really be represented solely by tensors of those size, since those tensors lack the information of where are the split points. I realize this representation is sufficient for the later manipulation, but this could be formulated more precisely.

With this in mind, I am inclined towards rejection, though my opinion can be swayed, given that my concerns are sufficiently addressed.

Minor comments:
- L52 missing -ed in interpretability-oriented
- L90 refers to $g$ that is not in the equation
- L130 incorrect article: "following *a*"
- L189 typo "s" in "leaf node*s* functions"
- L243 the "absolute difference of the matrix" should possibly be a sum or a max over matrix values or some norm
- L264 the density should probably be probability?
- L288 the j should probably not be a part of the iterators in the sums
- L377 "less comparable" is confusing, "perform worse" would be clearer
- L420 in $X_1 x X_2$, $\times$ would look better and would not be confusing.
- L479 The term "high-way" is confusing, a "multi-way" might be clearer.
- L852 "weather" instead of "whether"
- The used method would be much clearer if a pseudo-algorithm was provided. Esp. the effect pruning.
- In most tables with results, incl. Table 2, RMSE and AUC is confused, at least when comparing the values

**Questions:**

Main questions:
- Could the mehtod be extended to tree ensembles with linear functions in leaves instead of constants?
- The complexity analysis (L256+) ignores the number of iterations needed for convergence. Is it guaranteed to be negligible? What is the runtime, empirically?
- Why are there 10 main effects on the Friedman data, when the function contains just 5 inputs (x is 5-dimensional)?
- In Figure 3, if the feature importance is computed as sum of main effect and half of pair effects, why does the ordering of x1 and x2 different in effect and feature importances, when each should be added the same half of a single pair effect?
- In Figure 9b, why are there hr X holiday interactions twice?

Minor questions:
- How was $k=2$ selected (L344)?
- How is the threshold for candidates in FBEDk chosen?
- Is monotonicity guaranteed even after adding the effects extracted from multi-way interactions?

---

### Official Review · Reviewer_Y1fG · 2025-11-01

**Soundness:** 2
**Presentation:** 2
**Contribution:** 1
**Rating:** 0
**Confidence:** 4

**Summary:**

The authors propose a pipeline for building interpretable tree ensembles.
The pipeline consists of training shallow tree ensembles with "interpretability" constraints (e.g. on depth, monotonicity), decomposing the trained model via fANOVA into main effects and interactions, and pruning "trivial" effects using Lasso and a forward-backward selection. The authors show that XGBoost with depth 2 models can match or exceed EBM performance while being interpretable through visualization of 1D and 2D components.

**Strengths:**

The general topic of interpretable models is relevant (although the paper has a focus on tabular data and tree-based models only).

The authors provide clear exposition of how to map tree ensemble leaf nodes to fANOVA components.

The paper can be followed easily.

**Weaknesses:**

I am really unsure where exactly the contributions of the paper and the proposed pipeline lies (both with respect to theoretical insights / empirical insights).

Lou et al. 2013 (https://www.cs.cornell.edu/~yinlou/papers/lou-kdd13.pdf) already introduced GA2M models with main effects + pairwise interactions using shallow tree ensembles and showed these models often match full-complexity models empirically and they developed the FAST algorithm for efficient interaction detection which is more principled than post-hoc pruning pipeline proposed here.

fANOVA was introduced in detail in Hooker 2007 (https://www.tandfonline.com/doi/abs/10.1198/106186007X237892) and the decomposition in the present paper is only mechanical and purification was introduced in Lengerich 2020 (https://proceedings.mlr.press/v108/lengerich20a/lengerich20a.pdf).

Molnar et al. 2019 (https://arxiv.org/pdf/1904.03867) proposed model-agnostic complexity measures (NF, IAS, MEC) based on fANOVA and showed reducing these measures improves interpretability - especially for tree-based models.

Schneider et al. 2023 (https://dl.acm.org/doi/pdf/10.1145/3583131.3590380) benchmarked constrained shallow XGBoost across 20 datasets and showed XGBoost with depth 2 can outperform an EBM.
They also incorporated feature selection, interaction constraints, and monotonicity during training via multi-objective HPO and demonstrated the same core finding that shallow XGBoost achieves competitive accuracy with better interpretability.
Moreover, their approach is principled and results in obtaining a Pareto set of models that trade-off performance and interpretability / complexity.

The latter two papers are not cited by the authors although directly relevant.

Benchmarks are performed on only 4 small-to-medium datasets.
No statistical analyses are conducted.
There are no ablation studies, or sensitivity analysis of hyperparameters.
The paper also lacks scalability analysis showing computational cost as a function of n (number of observations) and p (number of features).
Competitors in benchmarks and experiments are only given by a GAM, EBM and GAMI-Net (although the authors mention plenty other related competitors, especially DNN approaches in the related work already).

**Questions:**

How does your approach differ from Lou et al.'s GA2M (with trees as base learners) beyond using post-hoc decomposition instead of training with FAST?

Can you comment on the relationship of your method to the works of Molnar et al. 2019 and Schneider et al. 2023?

Maybe I am missing something but given the related work mentioned above and under Weaknesses, can you clearly re-state what exactly the contribution of the paper is (beyond re-packaging existing ideas in a pipeline and using post-hoc pruning)?

Table 1 lists constraints as "contributions" but these are standard hyperparameters in XGB and investigated in any work concerned with interpretability (Molnar et al. 2019, Schneider et al. 2023). What is novel here?

---

### Meta-Review · Area_Chair_yKje · 2026-01-07

**Summary:**

The authors propose a pipeline for building interpretable tree ensembles. The pipeline consists of training shallow tree ensembles with "interpretability" constraints (e.g. on depth, monotonicity), decomposing the trained model via fANOVA into main effects and interactions, and pruning "trivial" effects using Lasso and a forward-backward selection.

While all reviewers agree that focus on interpretability for tabular prediction is important and timely, all of the reviewers suggest the paper should be rejected, although for different reasons. Y1fG suggests the paper is not original enough. tGa8 suggests the paper’s numerical results leave much to be desired (among others, “real-world data results are missing from the main body”, “performance is not shown to be superior to [other] methods” “Statistical significance was not tested”), and qYwq suggests the papers misses important baselines for comparison.

**Reviewer Concerns:**

The authors submitted no rebuttal, perhaps in anticipation of a rejection. While the concerns of Y1fG would be hard to alleviate, the concerns of tGa8 such as “real-world data results are missing from the main body”, while being advertised in the abstract, seem ok to address in a revision.

**Reviewer Scores:**

Considering there has been no rebuttal submitted, this question is moot.

---

### Decision · Program_Chairs · 2026-01-26

Reject